# Coping with COVID-19: Can a Sense of Coherence and Social Support Play a Protective Role in the Perception of COVID-19 by Polish Women Football Players? A Cross-Sectional Study

**DOI:** 10.3390/ijerph20136308

**Published:** 2023-07-06

**Authors:** Anna Ussorowska-Krokosz, Jan Blecharz, Malgorzata Siekanska, Monika Grygorowicz

**Affiliations:** 1Women’s Football Science Research Group, Department of Women’s Football, Polish Football Association, 02-366 Warsaw, Poland; 2Center for Mental Health in Sport, 81-342 Gdynia, Poland; 3Department of Psychology, University of Physical Education in Krakow, 31-571 Krakow, Poland; 4Department of Physiotherapy, Poznan University of Medical Sciences, 61-545 Poznan, Poland; 5Rehasport Clinic FIFA Medical Centre of Excellence, 60-201 Poznan, Poland

**Keywords:** pandemic, COVID-19, coping, female football players

## Abstract

Introduction: Athletes have demonstrated a significant increase in anxiety and stress connected to the COVID-19 pandemic. Drawing on the theory of salutogenesis, this study examined the relationship between the sense of coherence and social support and competitive elite-level athletes’ perception of the COVID-19 pandemic situation. Materials and Methods: The perceived secondary gains associated with the pandemic were analysed with a quasi-qualitative research methodology. The sample consisted of 204 competitive elite-level female football players aged 14–36 (M = 17.61, SD = 4.42) who completed the Sense of Coherence Questionnaire (SOC-29), the Berlin Social Support Scales (BSSS), and the researchers’ own questionnaire to assess the perception of the COVID-19 situation. Results: The results confirmed a relationship between a sense of coherence and coping better with the difficult situation caused by the coronavirus pandemic. No protective role of social support in the adaptive perception of the pandemic situation was identified. Conclusions: The data from the quasi-qualitative study testify to the use of the adversity of the pandemic to grow in many areas of female football players’ lives. The sense of coherence was a protective factor in mitigating the negative consequences of the COVID-19 pandemic situation.

## 1. Introduction

The situation caused by the COVID-19 pandemic undoubtedly has numerous consequences not only for physical but, above all, mental health [1,2,3,4]. The latest research confirms that uncertainty about one’s professional career and financial stability leads to a reduced quality of life and mental health problems [5,6].

Professional athletes are one of the groups who experienced numerous pandemic-related consequences [7,8,9]. The difficulties they had to face were associated with organisational, financial, and logistical changes [10,11]. The athletes had to stop their scheduled training overnight and cancel their participation in competitions. The limiting of physical activity, isolation from team members and the sporting community, as well as the lack of social support, had a negative impact on their psycho-physical condition [12]. In addition, the use of sports facilities was banned, and league and international competitions were suspended, which created another source of stress [8,13,14,15].

The pandemic situation undoubtedly triggered a number of psychological problems such as stress, anxiety, and depression [12,16]. What is particularly worrying is the fact that even before the COVID-19 pandemic, several alarming statistics on the mental health of professional athletes had been published, for example, by the European Federation of Sport Psychology (FEPSAC) [17] and the International Society of Sport Psychology (ISSP), showing that 16–34% of elite athletes had higher symptoms of psychological disorders [18,19].

Recent studies of elite soccer players have demonstrated a significant increase in anxiety and depression symptoms in comparison with studies carried out before the COVID-19 pandemic [20]. Unfortunately, there is still very little research concerning the pandemic situation and the ways of coping among female elite football players. Some players adapted more quickly to the new reality, using effective ways of coping, in comparison with others who experienced increased anxiety and symptoms of depression [21]. According to Antonovsky [22], the author of the concept of salutogenesis, we are constantly struggling with various types of stressors to which we do not have a ready-made, adequate adaptive response. Antonovsky’s model of salutogenesis proposes that health should be viewed as a continuum, rather than a dichotomy between sick and healthy. Rather than focusing solely on disease and illness, Antonovsky believes that health is a state of full physical, mental, and social well-being. The salutogenic approach emphasises the importance of understanding and enhancing the resources and strengths that promote health, rather than solely treating the deficits and weaknesses that lead to illness. Antonovsky’s model emphasises the importance of a sense of coherence (SOC), which refers to an individual’s ability to understand and manage life’s stressors. The stronger an individual’s SOC, the more likely they are to perceive stressors as manageable challenges, rather than threats to their well-being. Antonovsky believes that the effectiveness of coping is determined by three variables: the type and intensity of stressors, generalised resistance resources, and a sense of coherence [23].

Environmental stimuli which generate states of emotional tension and trigger adaptive responses are described as stressors. According to relational theory, the intensity of stressors is determined by an individual’s subjective evaluation rather than the objective properties of the person or the environment. The basic psychological mechanism, described by Lazarus [24], refers to a cognitive appraisal of threats seen as a process of evaluating, becoming aware of, and interpreting events. The individual considers the possibilities for taking action to eliminate the causes of stress or to achieve benefits [24].

Another variable determining the effectiveness of coping with stress is generalised resistance resources, described as the individual’s traits enabling them to avoid stressors or cope with the tension they generate [22,25]. The resources are associated with how the individual interprets stressors and copes with the stress they experience [23]. An example of a generalised resistance resource is social support, which can be classified according to the content and function performed in the course of social interaction. The most important kinds of support include emotional, informational, instrumental, tangible, and spiritual support [26]. Athletes receive such support from coaches, parents, or other athletes. In team sports, the awareness of being part of a group can be a source of perceived support [27]. Scholars agree that social ties largely determine an individual’s mental health and welfare [28,29,30]. Moreover, support plays a positive role not only in maintaining health but also in the recovery process [31]. Building a support network and close relationships facilitates balance in life [32] and adaptation to critical life events by mitigating the negative impact of stress and making it easier to cope with [33,34].

However, according to the salutogenesis model, it is the sense of coherence that plays a key role in coping effectively with stress [22]. The sense of coherence is a global orientation that expresses the extent to which an individual feels confident that (a) the stimuli coming from the internal and external environment are structured, predictable, and explicable; (b) he or she has the resources to cope with the demands posed by these stimuli; (c) these demands are a challenge worthy of effort and engagement. The construct comprises three elements: comprehensibility, manageability, and meaningfulness. Comprehensibility denotes the degree to which the surrounding world is perceived by an individual as orderly and coherent. Cognitive control of the environment is the essence of comprehensibility. Manageability is linked to a belief in having sufficient resources and possibilities to cope with unfavourable circumstances or demands of a given situation. Awareness of existing resources, not only one’s own but also those of family and friends, is at the heart of manageability. Both comprehensibility and manageability are linked to the cognitive aspect concerning a given individual’s beliefs. The third element of the sense of coherence—meaningfulness—has a motivational dimension of the construct. Meaningfulness in the context of salutogenesis refers to a sense of purpose and significance in life. It is the feeling that one’s life has value and that their everyday actions are contributing to a greater purpose. Research has demonstrated a positive link between the sense of coherence and mental health [35,36], as well as a negative link between the sense of coherence and anxiety, depression, and negative emotions [37,38]. The sense of coherence not only determines mental health but is also a factor enabling athletes to achieve excellent results in competitive sports [39,40]. Although SOC has been proven to be a predictor of whether athletes can cope better with the COVID-19 pandemic [41,42], it hasn’t been investigated in female soccer players.

The salutogenesis model makes it possible to explain and understand the effectiveness of coping with stress. Individuals with a high sense of coherence and generalised resistance resources see stressful situations as less threatening, and as challenges they are able to cope with. Such an interpretation enables them to see, even in the most difficult situations, some secondary gains, perceived benefits [43], positive changes in outlook [44], post-traumatic growth [45], stress-related growth [46], and adversarial growth [47]. The phenomenon has been widely observed in the context of athletes recovering from a physical injury who, despite their negative experiences, notice the benefits associated with the situation [48,49]. Conversely, the isolation caused by the COVID-19 pandemic has received little attention so far.

Taking into account the above considerations and the scarcity of studies on the relationship between the sense of coherence and social support on the one hand, and competitive elite-level athletes’ perception of the COVID-19 pandemic situation on the other, the following hypotheses were formulated: H1—the higher the sense of coherence, the more adaptive approach to the situation; H2—the higher the social support, the more adaptive approach to the situation. In addition, responses concerning perceived secondary gains associated with the pandemic were analysed in line with the quasi-qualitative research methodology.

## 2. Materials and Methods

### 2.1. Design, Procedures, and Participants

The study was designed as a cross-sectional online survey which took about 25 min to fill in. It was a part of the Polish Football Association (PFA) project, dedicated to supporting football athletes during the pandemic of COVID-19, and it was fully assisted by the PFA in terms of organizational aspects. This provided access to the coaches of national teams (*n* = 4) as well as all the teams registered in the highest women’s division, the Extra League (Ekstraliga) (*n* = 12), First Division (*n* = 12) Second Division (*n* = 24), and the youth teams U-17 (*n* = 20) taking part in female football competitions organized by the PFA. All the clubs received an invitation from the study coordinator (MG), who is the Head of the Women’s Football Science Research Group in the Department of Women’s Football in the PFA. The clubs’ authorities were asked to point out the person in each club who could act as the first contact for this project. Then, the first author (AU-K) sent the project description to these contacts clarifying the objective of the study, the timeline, and the sampling process in detail. Finally, a contact person from each club provided the participants with the link to the online questionnaire. Consent in writing was obtained from each participant and, in the case of minors, the consent of their legal guardians. The only inclusion criterion for the study was playing football at the female competitive elite level in a club registered in the PFA. The research was carried out in accordance with the Helsinki Declaration. During an introductory meeting, the participants were introduced to the idea of the study, and the ethical aspects were explained (voluntary participation, confidentiality in data treatment, and presentation). Additionally, written consent from each participant was obtained. The survey was conducted at the end of May 2021, a month after the end of the most restrictive lockdown in Poland (March–April 2021). During this time, clubs stopped training, and national and international competitions were suspended. All the surveyed clubs limited their support to sending training schedules in electronic form during the pandemic.

### 2.2. Measures

Each participant completed an online survey, which consisted of questions regarding their personal profile (club, level of play, age, years of training, athlete’s status). In the next part, the participants responded to the statements that applied to the COVID-19 situation. The study included the researchers’ own questionnaire assessing the perception of the COVID-19 situation, the Sense of Coherence Questionnaire SOC-29 [50], and the Berlin Social Support Scales BSSS [51].

The researchers’ survey comprised 15 questions, 7 of which were open questions. The questions concerned how players interpreted the difficult situation, whether they reformulated their sports goals or whether they perceived any benefits related to the pandemic, and what needs and expectations they held (see Appendix A). 

A Polish adaptation and validation [50] of the Sense of Coherence Questionnaire [52] was used. The questionnaire consists of 29 questions and comprises three subscales: comprehensibility, manageability, and meaningfulness, measuring the three components of the sense of coherence. The reliability coefficients of this tool range between Cronbach’s alpha of 0.82 and 0.95 [23]. The studies conducted on the Polish population indicate a reliability coefficient between Cronbach’s alpha of 0.85 and 0.88 [25].

A Polish adaptation and validation [53] of the Berlin Social Support Scales [51] was also used. This questionnaire is used to measure social support and comprises 6 scales: perceived support, actually received support, need for support, support seeking, protective buffering support, and support provided by a loved one. The tool comprises 38 questions. The reliability coefficients of this tool range from Cronbach’s alpha of 0.83 to 0.86 [51].

### 2.3. Statistical Analyses

Statistical analyses were conducted using the Polish version of STATISTICA 12 (TIBCO, Palo Alto, CA, USA), the Student’s *t*-test, and multiple regression. The analyses were carried out in two stages. First, a statistical description of the analysed variables—the sense of coherence, and social support—was performed, followed by the analysis of participants’ responses to questions concerning the assessment of the players’ situation during the first COVID-19 lockdown in 2020. In the second stage, the researchers used multiple regression to test the first and second hypotheses.

Since the research was not qualitative per se, the answers to one question, i.e., the most important one in the context of this study, were analysed in a quasi-qualitative way. The question concerned the presence and type of secondary gains perceived by the participants during the COVID-19 pandemic. The objective of the quasi-qualitative data analysis was to interpret the meanings attributed to the subjects observed by the researchers [54]. The coding process was based on the thematic analysis framework, where topics and themes are identified by experts in the field. The quasi-qualitative data analysis consisted of three parts: data coding, looking for and establishing links between concepts, and interpreting the data. Specific procedures were performed at all the above stages by 5 independent competent judges.

## 3. Results

### 3.1. Participants

The study involved competitive elite-level female football players, *N* = 204, aged between *min* = 14 and *max* = 36 years (*M* = 17.78; *SD* = 4.40). The participants were the players of the national senior team and three national youth teams. In total, they represented 54 football clubs, but 8 players did not report the name of their club. Most of the participants played for Polish clubs (93.92%), and the remaining 6.02%, represented foreign clubs. In the sample, 49.56% of the footballers played for both the league clubs and the national team, while 50.44% played for the league clubs only. According to the classification proposed by Swan, Moran, and Piggott [55], which divides the population of elite athletes into semi-elite, competitive elite, successful elite, and world-class elite, the majority of the female football players in the study were at the competitive elite level. Three players among the participants could be classified as being part of the successful elite. The classification is based on an assessment of the player’s achievements in their discipline (within sports comparison), as well as the popularity of the discipline in question (between sports comparison). A significant majority of the female football players combined sport with education (83.91%). Only 10.00% responded that football was their only occupation, while 6.09% combined sport with gainful employment.

### 3.2. Quantitative Analysis

The first step in the analysis was to determine the level of the participants’ sense of coherence. The results showed interesting differences between females playing only for league clubs and those who also played for the national team. In all dimensions of coherence, the national team players obtained statistically significantly higher results than league football players (Table 1).

As the analysis of the Berlin Social Support Scales did not reveal any differences between the league and national team football players, the results are presented without a division into groups (Table 2).

No differences were found between the national team and the league club players in the perception of the COVID-19 pandemic situation (Table 3).

The next step in the analyses consisted of testing hypotheses 1 and 2 using multiple regression. When it comes to hypothesis 1, the independent variables were dimensions of the sense of coherence (3), while the dependent variable was the assessment of the perception of the COVID-related situation (6 × 1). Detailed results of the regression analysis are presented in Table 4.

Multiple regression analysis was also used to test hypothesis 2, in which the dependent variable was the assessment of the perception of the COVID-related situation, and the independent variables were the various dimensions of support. The regression analysis demonstrated that none of the support models were statistically significant. In addition, no dimension of support was found to be a significant predictor of the perception of the COVID-19 situation.

### 3.3. Quasi-Qualitative Analysis

As emphasised earlier, this study does not meet the criteria of a qualitative study; however, the most important question in the context of this study was analysed using the quasi-qualitative methodology. The objective of the quasi-qualitative analysis was to formulate conceptual constructs, and/or theoretical and hypothetical constructs.

The questions concerned the presence and type of secondary gains perceived by football players during the COVID-19 pandemic. In particular, they aimed to better understand the perceptions of the COVID situation by the participants. In addition, the participants had the opportunity to answer open-ended questions about their objectives and provide an assessment of their condition.

The data coding (concept formulation) stage followed procedures of open coding, case summary, focused category coding, and category saturation [56]. In order to determine the relationships between concepts, the researchers used procedures of initial theoretical memos concerning specific links between phenomena and corresponding concepts. The final stage of the analysis consisted of data interpretation (theorizing) to create diagrams integrating and representing the coded empirical data summarising the theoretical memos [57]. Five independent judges participated in the data analysis process.

Secondary gains observed by the football players (Figure 1) were associated with a range of variables. However, an analysis of the data captured some extremely important processes in the participants’ responses. Two elements repeatedly pointed out among the benefits of the COVID-19 pandemic were self-reflection and an opportunity to step back from daily functioning in sports. The period of isolation was a time during which the players could think of and redefine their priorities. Reflection on their lives enabled them to see and appreciate the resources they had, especially their relationships with other people. Looking at their lives from a different perspective also led them to another observation—the need to develop not only in football but also in other areas not connected with the sport.

#### 3.3.1. Sport

As far as secondary gains are concerned, the participants pointed primarily to the opportunity for development as an athlete. They used the period of isolation and suspension of competitions to improve their individual technical and motor skills. Some of the participants also delved into tactical aspects, analysing football matches. The players perceived the COVID-19 situation as an opportunity to comprehensively take care of their fitness. The activities they engaged in related to the development of football skills, but also a broadly defined healthy lifestyle, such as sleep quality and a balanced diet.

#### 3.3.2. Education

The isolation caused by the COVID-19 pandemic created an opportunity for football players to self-develop. Among the secondary gains, the players pointed to the possibility of catching up with school and university work. The period also posed an opportunity to take part in all kinds of online training courses or studies. The participants indicated that education-related activities comprised the ones related to football, but also other topics not associated with it. In addition, self-development through reading books was considered as the secondary gain resulting from isolation.

#### 3.3.3. Self

The suspension of training and league matches meant that the participants had more free time at their disposal. For elite female football players, whose days are filled with training sessions and numerous other activities, this was undoubtedly a new situation. It was also an opportunity to take a step back and look at themselves from a different perspective. The participants pointed to the possibility of self-reflection and defining new priorities in their lives as secondary gains. In addition, they indicated that the period of isolation was a time during which they discovered new passions and interests. Some of them worked during the pandemic on projects relating to their functioning after the end of their careers as professional football players.

#### 3.3.4. Relationships

Elite sports include a substantial number of training sessions, matches, tournaments, and training camps. Such a way of functioning is not conducive to building lasting relationships, especially outside football. Perhaps this is why the participants pointed to the possibility of spending time with their families and friends as another secondary gain resulting from the isolation. Not only did the participants appreciate the time spent with their families and friends, but they were also grateful for the social support they received from them. In addition to maintaining relationships with their loved ones, some participants also appreciated the possibility of building new relationships with people from outside the sports environment.

## 4. Discussion

The study confirmed the first hypothesis concerning a relationship between the sense of coherence and the ability to cope better with the difficult situation caused by the coronavirus pandemic.

Undoubtedly, the most interesting result was the one indicating that manageability—a dimension of the sense of coherence—was a protective factor in the assessment of difficulty during isolation. Similarly, research conducted by Szczypińska, Samełko, and Guszkowska [42] demonstrated that manageability in Olympic-level athletes negatively correlated with negative mood dimensions during the pandemic (such as confusion, fatigue, or depression) and positively correlated with vigour. Manageability could, in such a case, be responsible for task orientation in a difficult situation and, consequently, enable the players to better organize their activities. This finding is in line with the concept proposed by Orlick [58,59], who specified two zones in the lives of athletes—gold and green. The former is related to sports activities such as tactical, technical, physical, and mental preparation, as well as competition. The latter refers to life outside sports, i.e., family, friends, and leisure time. Effective functioning in the gold zone is determined by a balance between the two zones.

Interestingly, comprehensibility was not found to be a predictor of treating the COVID-19 situation as a challenge or noticing some lockdown-related benefits. There are still relatively few studies exploring the sense of coherence in athletes during the pandemic, which is why explaining the phenomenon based on reports is difficult. Presumably, the situation caused by a global crisis, with all the related changes in behaviour and lifestyle of a majority of people, was hard to comprehend. Experts in epidemiology, virology, and social sciences faced the challenge of explaining the situation in an intelligible way for everyone. Furthermore, conspiracy theories or opinions simplistically explaining the COVID-19 situation proliferated online, in particular, on social media. Such explanations may create an illusion of comprehension, but it would be very risky to say that they contributed to an increase in welfare, a sense of control, and, consequently, the ability to better cope with adversity. In the study of Szczypińska et al. [60], comprehensibility correlated positively with negative dimensions of mood (confusion and depression), suggesting it was not so much comprehension of the COVID-19 situation, but the acceptance of the discomfort associated with incomprehension. Similar conclusions were drawn by Gupta and McCarthy [61], who indicated that the acceptance of the incongruence of the situation was a protective factor against rumination-like recurring negative thoughts.

The study, however, did not confirm the second hypothesis—the protective role of social support in the adaptive perception of the pandemic situation. Most studies suggest that social support in athletes’ lives can protect them against the negative consequences of chronic stress, burn-out, and the risk of mental disorders [27,62,63,64,65]. However, there is one major difference between the adversity experienced by the players before (physical injury, fatigue, overtraining, or consequences of defeats and the fight to stay in the league) and during the pandemic situation. In the case of sporting adversities, the players were able to rely on the support of people not burdened with the difficult situation (family, friends). In the case of the pandemic, on the other hand, both the football players and their entire support network experienced the negative effects of isolation and pandemic-related restrictions. The COVID-19 situation can certainly be classified as a non-normative change in the context of sporting career development [8]. The development of a career in sport involves going through its successive stages, among which normative changes are distinguished that are relatively predictable and include, for example, the transition from junior to senior level, and non-normative changes that the athlete cannot predict (e.g., the COVID-19 pandemic) [66,67]. Athletes find it difficult to prepare for non-normative changes, which are low in their predictability [68]. The perception of a difficult situation and the athlete’s reaction may be determined by, for example, perceived control of the situation, motivation to participate in sport, coping skills, and available support [69,70].

Another reason for the lack of a protective role of support may be a mismatch between the support received and the real needs and expectations of the players. Perceived support shares only a small portion of the variance with the support received [65]. Research confirmed that, in comparison with the support received, perceived support was a better predictor of both psychological well-being and the ability to cope with stress more effectively [65,71].

The findings from the qualitative part of the study concerning the use of the adversity of the pandemic to grow in many areas of female football players’ lives were in line with the observations from a meta-analysis conducted by Howells, Sarkar, and Fletcher [72]. Accordingly, athletes had the potential to benefit (secondary gains) from adverse situations such as physical injuries, illnesses, or problems within families. Consequently, the authors found that the most frequent benefits were personal and social growth, the growth of awareness, the ability to change perspective or work on technical skills, and engagement in rehabilitation. However, it is worth noting that the meta-analysis in question was conducted before the outbreak of the pandemic in 2020; thus, it did not include athletes’ way of reacting to this type of adversity. The issue of secondary gains was also investigated in a qualitative study conducted by Woodford and Bussey [73]. Fourteen elite athletes who participated in this study reported that time away from sports encouraged them to reflect on their athletic identity and to make life changes [73]. The results obtained in the presented study and the cited literature suggest that flexibility and acceptance of discomfort are key competences in the current world of sports.

### Strengths, Limitations, and Future Directions

The strengths of the study undoubtedly include the fact that the research sample comprised a significant majority of professional female football players in Poland, competing on both the league and national team levels. In addition, the strength of the study is determined by its nature of mixed-method research—combining features of quasi-qualitative analysis and statistically processed quantitative data.

The main limitation of the study is its cross-sectional nature. The survey was carried out a month after the easing of the pandemic restrictions and hence represented the retrospective nature of the participant’s assessment of the threat and discomfort. This limitation was caused by the unpredictable dynamics of changes in the development of the pandemic. In addition, variables such as the levels of physical activity and the places athletes spent their time in isolation were not scrutinized.

The findings concern Polish football players only, and it would be relevant to conduct research applying intercultural comparisons [74]. Such a design would facilitate the generalisation of the results encompassing a much larger group of female football athletes. Furthermore, the results of this study showed an important role of the sense of coherence in coping with adversity; therefore, in future research, it would be relevant to test the effectiveness of interventions based on promoting and building such a sense. In addition, given how difficult it is to predict the development of the pandemic, it would be advisable to conduct a longitudinal study monitoring changes in levels of perceived stress, taking into account variables such as the sense of coherence, social support, and subjective gains from the adversity in question.

## 5. Conclusions

The results of the study confirm that female football players with a higher sense of coherence were able to adapt more quickly to the COVID-19 pandemic situation. Moreover, they saw it as a challenge they were able to cope with. Thus, the sense of coherence was a protective factor mitigating the negative consequences of isolation. On the other hand, the protective function of social support was not confirmed. The availability of social support was not linked to the perception of the COVID-19 situation. Social support is a generalised resistance resource, which, however, was not linked to the type of interpretation of the female football players’ situation. Despite the closure of sports facilities and suspension of league competitions, the participants were able to see some secondary gains in the crisis. The period of isolation was a time during which the players were able to take a step back and redefine their priorities. Reflection on their lives enabled them to appreciate the resources they had, in particular, their relationships with other people. The participants noticed that they needed to develop not only in football but also outside sports.

## Figures and Tables

**Figure 1 ijerph-20-06308-f001:**
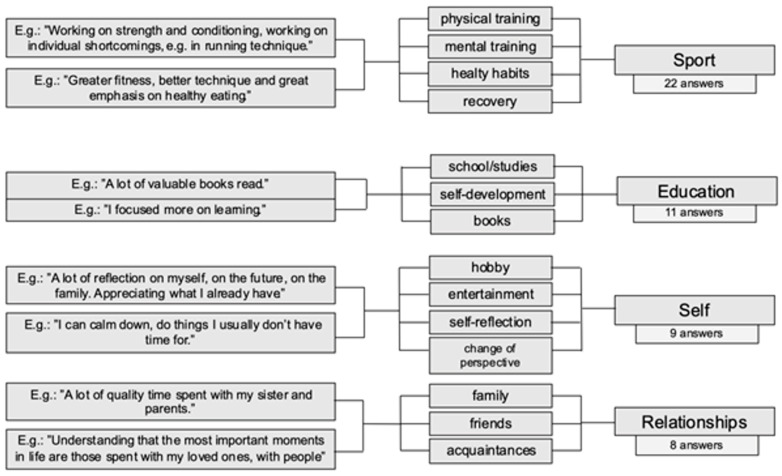
Graphic representation of perceived secondary gains.

**Table 1 ijerph-20-06308-t001:** Results of the Sense of Coherence.

	National Team (*n*) and League (l)				
	*M* (l)	*M* (*n*)	*SD* (l)	*SD* (*n*)	*p*
comprehensibility	42.27	46.06	9.10	9.94	0.006
manageability	48.28	52.02	9.17	8.58	0.003
meaningfulness	37.45	40.68	5.91	5.20	<0.001

**Table 2 ijerph-20-06308-t002:** Results of the Social Support.

Support Scale	*M*	*min*	*max*	*SD*
perceived available social support	14.45	8.00	16.00	1.53
perceived available instrumental support	15.01	10.00	16.00	1.38
need for support	12.72	4.00	16.00	1.56
support seeking	15.33	5.00	20.00	2.48
emotional support	31.57	18.00	36.00	3.89
instrumental support	10.99	7.00	12.00	1.19
informational support	7.07	2.00	8.00	1.20

**Table 3 ijerph-20-06308-t003:** Perception of COVID-19 Pandemic by Players.

Perception of the COVID-19 Pandemic	*M*	*min*	*max*	*SD*
the current pandemic-linked situation is difficult for me	3.34	1.00	5.00	1.01
the pandemic prevented me from achieving my goals	3.23	1.00	5.00	1.11
I have effective ways of enabling me to cope	3.46	1.00	5.00	0.82
I think I can use the current situation in a positive way	3.47	1.00	5.00	0.87
I have modified my goals because of the current situation	2.66	1.00	5.00	0.92
I see benefits from the time spent during the pandemic	3.25	1.00	5.00	1.00

**Table 4 ijerph-20-06308-t004:** Results of Multiple Regression Analysis—the Sense of Coherence and Perception of the COVID-19 Pandemic.

	β [*p*]			Multiple Regression		
	Comprehensibility	Manageability	Meaningfulness	Percentage of Variance Explained	*F*	*p*
perceiving COVID-19 as a difficult situation	−0.02	−0.26 [0.035]	0.13	5%	3.29	0.027
being prevented from achieving goals	0.03	−0.11	−0.02	1%	0.65	0.584
having effective ways of coping	0.15	0.09	0.18 [0.050]	12%	8.93	<0.001
using COVID-19 in a positive manner	0.12	−0.12	0.19 [0.049]	4%	2.59	0.053
need to modify goals	0.03	−0.08	−0.03	1%	0.48	0.691
seeing pandemic-related benefits	0.08	−0.15	0.18 [0.049]	2%	1.62	0.186

β [*p*]—regression coefficient, *F*—F-test value, *p*—level of significance.

## Data Availability

Data supporting reported results can be achieved from the corresponding author.

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
