# Peer review of "Coping with COVID-19: Can a Sense of Coherence and Social Support Play a Protective Role in the Perception of COVID-19 by Polish Women Football Players? A Cross-Sectional Study"

_ijerph, 2023, doi:10.3390/ijerph20136308_

Round 1
Reviewer 1 Report
Review of manuscript ijerph-2288663 “Coping with Covid 19: Can a sense of coherence and social support play a protective role in the perception of Covid-19 in Polish women soccer players? a cross-sectional study”
This is a study with 204 elite level Polish female soccer players after a lockdown in 2020 during the Covid19 pandemic. Quantitative variables were sense of coherence (three subscales), social support (six subscales) and six items (five point Likert scales) concerning possible side effects of the pandemic. In addition, participants were asked for perceived secondary gains of the pandemic. The answers to this question were structured into four groups – sport, education, self, relationships (Figure 1).
Comments
Even though soccer is a popular sport in many countries and also interest in female participation is growing, research with female soccer players is very scarce. Therefore, the current study deserves attention in itself. On the other hand, the manuscript reveals some flaws which may be considered in a revision. In the following I make some suggestions and ask some questions in order to improve the manuscript.
First of all, please make a thorough language check in order to improve the comprehensibility.
In the following, I proceed with my comments depending on the line numbers.
l. 69: Lazarus
l. 70 to 75: you are speaking of primary and secondary appraisal, and in between you are speaking of cognitive = primary and emotional = secondary. This is confusing and in addition, it does not correspond to the stress model of Lazarus. Please consider either to delete lines 70 to 74, or to explain the Lazarus model in greater and (correct) detail.
l.81: resistance
l. 195; l. 242: please avoid talking about verifying hypotheses. Instead, prefer to say “test” or “investigate” hypotheses.
l. 202-205: What is meant with “quasi-qualitative”; In lines 271-277 you repeat the text and apart from that, I have problems to understand what you with these explanations. Unfortunately, the references no. 54 and 56 are in Polish which makes it difficult to check the literature for a better understanding of your procedure. In Figure 1, you report the results and mention numbers. Is this the number of participants who answered that question, or is it the number of comments per category? How important is this part of your study? Can you imagine to skip it completely, because there is no connection to the quantitative results.
Table 1: In the text you mention differences between athletes, but the table does not correspond to that. There are only results from one group (or the whole sample?).
Table 4: Which kind of regression analysis did you make? And how much variance is explained by the independent variables? The p-values are quite close to .05, but perhaps the explained variance is higher?
l. 416-421: in the limitations you mention that the study is correlational. How does that influence your hypothesis and the conclusions/interpretations? It would also be interesting to know what the athletes were doing in the phase of lock-down. Did you gather any information on that?
Author Response
Dear Reviewer,
Thank you for your effort and for preparing the revision. We corrected the current version of the manuscript according to your suggestions. I hope all our explanations will be clear to you.

Reviewer 2 Report
First of all, thank you for the opportunity to review this interesting manuscript. This study touches on very important aspects of the mental health of the female representatives of team sports in the period of confinement and the pandemic. However, I have some questions and would like to suggest some changes in different parts of the manuscript. Here are my comments:
Introduction
In line 5o the authors state that before the Covid-19 pandemic there were "alarming statistics" for the mental health of professional athletes. I would recommend giving accurate statistics and highlighting this problem by showing what percentage of athletes suffer from affective problems.
The authors write in line 77: "The second variable..." but do not mention "first variable", is this "cognitive appraisal" or "intensity of the stressor"? This needs to be clarified in the text.
I would recommend to connect all the salutogenesis model of Antonovsky and place it as one paragraph.
In line 106 the authors do not pay much attention to the description of the variable "meaningfulness". This should be developed. (Especially since it is a very significant predictor of coping)
In line 110-111 the authors write a sentence sounding strong with hard evidence on SOC and coping in COVID-19. This sentence makes the current study less valuable (SOC has already been shown to be effective in coping with stress). Therefore, I would recommend to add a sentence here that would underline the uniqueness of the current study, that SOC was not investigated in female soccer players, which makes it so worthwhile.
Materials and Methods
In line 139 there should be Extra League instead of Ekstraliga, which is a Polish word.
The procedure could be supported with the flow chart.
Please remove the text from lines 172-177 which explains the sense of coherence already described in the introduction section.
Please give a developed description of the SOC questionnaire and the Berlin Social Support Scale. Give examples of one statement for each scale measured. Also describe the type of response scale for each method (Likert scale 1-5).
Please provide a statistical analysis with information on how the normality of the distribution of the study variables was developed.
Results
I would recommend adding the table presenting the descriptive statistics. This table could also be mentioned in the text where procedures and participants are described (Materials and Methods).
Please provide the results with specific, accurate statistical results. In the description of the significant result from Table 1, there is no information about the p-value. If this is the significant result, I would like to know how strong it was.
The same is true for the description of the regression. Please support the description with specific numbers on how the SOC predicts (explains, influences) the perception of COVID, in what percentage, etc. On line 246, the authors write: "The regression model was significant in two cases", please provide details.
In lines 289-294 there is a fit with Orlick's concept. In my opinion, such parts should be left out and put up for discussion.
3.3.1, 3.3.2, 3.3.3., 3.3.4 I would write in italics, which follows the rules of the entire manuscript.
Discussion.
I really admire the flow of this part of the text. However, in some places it is hard to understand when the authors are describing the current study results and the other researchers. For example, in lines 340-347, the authors start with the sentence about studies on Taekwon-Do athletes, but there is no citation. Then they probably describe their results and then go back to the results of Guszkowska et al. I would recommend to explain the current results first and then compare them with others. This would make the discussion more readable for the reader.
I would not include statistical numbers in the Discussion (lines 342, 343, 392).
Lines 386,387 I would recommend correcting this sentence.
The paragraph on qualitative study discussion (starting from line 396) - I would bring back some of the most interesting results from the Results section here to underline how the athletes' flexibility towards difficult and unpredictable situation of Covid helps them to cope with it.
Author Response

(The authors gave the same response as above.)

Reviewer 3 Report
Thanks for submitting the manuscript. However, the following issues significantly undermined the quality of the manuscript, which I think the current form of the manuscript is not suitable for publication:
1) The use of quasi-qualitative research methodology on 204 recruited samples is not desirable. The authors did not further clarify the coding process and the analytical framework.
2) In the literature review section, the authors have been clarified various variables. To what extent they are related to the underlining research questions and the hypotheses? The literature also suggested that there are linkage between those variables, what are the new insights of this study?
3) Did the measures SOC-29 and BSSS have been validated in the context of Poland?
4) Another important issue is that the study was not endorsed by the ethical review committee (p. 3). I understand that the Polish law may not require to do so, but IJERPH is an international journal, authors still need to abide to the Committee on Publication Ethics (COPE) guidelines.
5) The multivariate analysis is a bit lack of scientific rigour. There are no controlling variables in the models. Are there any mediating effects between the variables?
6) Another concern is the level of the recruited women soccer players. Combining the national teams with the semi-professional and junior team members in the same pool is quite problematic.
Author Response

(The authors gave the same response as above.)

Round 2
Reviewer 1 Report
Thanks for your revision. Please make a language check of your manuscript
Author Response
Dear Reviewer,
We would like to thank the reviewers for all their comments.
Thank you for the opportunity to improve and resubmit our manuscript. In order to fully respond to the comments of the reviewers, we carefully analyzed all suggestions.
We enclose a certificate drawn up by a professional English translator.
Comments on points 1, 2 and 6, formulated by Reviewer 3, are not clear.
To the best of our knowledge, we have made every effort to address all comments from reviewers.
Reviewer 3 Report
Thanks for submitting the revised manuscript. However, the authors still failed to fully address the issues related to the research design (point number 1 and 6) and data analysis methods (point number 2). In view of the above problems, I do not think the current form of manuscript is suitable for publication.
Author Response

(The authors gave the same response as above.)
